# The Impact of COVID-19 Health and Safety Measures on the Self-Reported Exercise Behaviors and Mental Health of College Students

**DOI:** 10.3390/ijerph20247163

**Published:** 2023-12-11

**Authors:** Justin A. DeBlauw, Mary Stenson, Astrid Mel, Daniela German, Aaron Jaggernauth, Brian Lora, Noa Schabes, Raymani Walker, Farouq Yusuf, Stephen J. Ives

**Affiliations:** 1Health and Human Physiological Sciences, Skidmore College, Saratoga Springs, NY 12866, USAblora@skidmore.edu (B.L.);; 2Applied Human Sciences, University of Minnesota-Duluth, Duluth, MN 55812, USA; 3Exercise Science, Mercy University, Dobbs Ferry, NY 10522, USA; amel@mercy.edu

**Keywords:** physical activity, mental health, college

## Abstract

The public-health restrictions (e.g., remote learning, restricted access to facilities and dining halls) put in place by colleges to reduce the spread of COVID-19 resulted in forced isolation and modifications to health-related behaviors. The restrictions and uncertainty associated with COVID-19 may have exacerbated the challenges of meeting exercise recommendations and mental-health concerns. The purpose of this study was to assess the impact of restrictions on students’ exercise habits and their levels of anxiety, stress, and depression. Five-hundred and forty students completed a 29-question survey on individual demographics, living arrangements, exercise, sleep, diet, and mental health. Significant changes in weekly days of exercise and intensity were reported. Increases in anxiety, stress, and depression were reported. The two most frequently reported changes in exercise behavior were an increase in minutes of aerobic training (5%) and a combination of reduced minutes of aerobic and resistance training (3.9%), which could be reflective of an individual’s ability and/or desire to maintain exercise behavior during the restrictions. Alternatively, for those students who reduced their exercise habits, aerobic training (11%) was the mode that suffered the most. Demographic factors such as ethnicity, regional residence, and gender were found to have significant effects on stress, anxiety, and depression. Amidst pandemics and future health emergencies, colleges should prioritize establishing opportunities for students to exercise, helping them meet physical activity recommendations and combat mental-health issues.

## 1. Introduction

Due to the rapid global spread of Severe Acute Respiratory System Coronavirus 2 (SARS-CoV-2 virus), with health effects ranging from asymptomatic to severe, including death, the World Health Organization (WHO) declared a global pandemic on 11 March 2020 [1]. In response, state governments, colleges, and universities enacted local restrictions, including stay-home orders and limits on outdoor activities that limited access to fitness centers, trails, and parks. While these restrictions may be effective in reducing the spread of viruses, they may also disrupt engagement in routine physical activity (PA) and exercise, while also being linked to symptoms of depression and anxiety [2]. Regular engagement in PA and exercise offers numerous benefits, including improvements in cardiovascular health, strengthened immunity, and reduced risk of chronic diseases, which can improve the body’s response to vaccines and may reduce the severity of symptoms of the coronavirus disease 2019 (COVID-19) [3,4,5,6]. As the COVID-19 pandemic continues, in addition to the likely possibility of future pandemics, it is vital to understand the impact of public-health/safety restrictions on individual physical and mental health.

It has widely been reported that mandated restrictions (e.g., restricted access to public spaces and closure of fitness facilities) in response to the spread of COVID-19 resulted in an increase in sedentary behavior and a decrease in PA and exercise in the general population [7,8]. However, due to the unique nature of universities and colleges, there may be differences in PA and exercise between the general population and university students. Surveys of university students reported a 34% reduction in mild PA following the cancellation of in-person instruction, a 14% increase in weekly sitting time, along with 49% not meeting the moderate-to-vigorous physical activity (MVPA) guidelines [9,10,11]. In contrast, these aforementioned observations challenge the results of surveys of the general population, which reported increases in the total minutes of PA [12,13]. The most notable change in PA in the general population was in those individuals who reported being usually active before restrictions, reporting a reduction in MVPA and a reduction in overall PA during restrictions [8,9,14]. Additionally, for those who maintained their levels of PA, there has been an observed mode shift such as decreased resistance training and increased activities of daily living [13,15]. There has been relatively little study of college-aged students with regards to alterations in their physical activity patterns as a result of the COVID-19 pandemic mitigation strategies, which may have varied across institutions, states, and countries.

While physical activity results in notable changes, the impact on mental health is equally of concern. The role of PA and exercise on mental health could easily be overlooked while attempting to reduce the spread of COVID-19; however, PA and exercise can play an important role in managing mental-health symptoms such as stress and anxiety resulting from COVID-19 mitigation measures [8,16,17]. Perhaps especially vulnerable to a decline in mental health are university and college students who have been challenged with the stress of online and social/life adjustments in response to COVID-19 restrictions [18]. A survey of university students found that 71% of respondents had increased levels of stress and anxiety due to the outbreak of COVID-19 [19]. Additionally, the Centers of Disease Control and Prevention (CDC) reported that 63% of college-aged adults had elevated levels of depression or anxiety between May and June 2020 [20]. Due to the positive impact of PA and exercise on anxiety, stress, and depression [21,22], it was recommended as a therapeutic treatment for physical and mental health during the COVID-19 pandemic [21]. Despite the documented effects of the pandemic on the general population, there is limited research focused specifically on the intersection of exercise and mental health in college students during the COVID-19 restrictions.

The primary aim of this study was to determine if college students experienced changes in self-reported exercise behaviors (e.g., days, intensity, and minutes) or in mental health (e.g., anxiety, stress, and depression) during the COVID-19-related restrictions. Additionally, we aimed to examine the effect of living situations (e.g., state of permanent residence and roommate status) and demographics on exercise and mental health during COVID-19-related restrictions.

## 2. Materials and Methods

### 2.1. Recruitment and General Procedures

In February 2021, a convenience sample of participants was recruited through the use of email announcements; social media platforms; and distribution by Dean of Students offices, Student affairs offices, and individual faculty. Participants were primarily recruited from Skidmore College, higher education institutions in the New York 6 consortium (Union College, Hamilton College, Colgate College, St. Lawrence University, Hobart and William Smith Colleges), and institutions beyond the NY6. Participants were required to be at least 18 years old and a full-time student at an undergraduate institution. No other inclusion or exclusion criteria were used. Students consented to the study prior to participating in the survey. The study and survey were approved by the local Institutional Review Board at Skidmore College (IRB#2102-944).

### 2.2. Survey

Data were collected between (1 February 2021–10 March 2023) via an online survey administered through Qualtrics Online Survey Software (Provo, UT, USA). The survey consisted of 29 individual questions using a series of yes–no, multiple choice, or Likert-scale questions. The survey consisted of three main sections, individual demographics, living arrangements, and health behavior. The health behavior section asked respondents about their exercise behavior (frequency, intensity, and duration), sleep, diet, and mental health (stress, depression, and anxiety) before and after the start of the COVID-19 pandemic when colleges/universities closed. The exercise responses were multiple choice, frequency ranged from 1 day to 7 days a week, intensity ranged from without an intensity to maximal intensity and duration ranged from 0 min to more than 120 min. Reponses for self-reported changes in mental health captured via Likert-scale questions ranged from (1) decreased significantly to (5) increased significantly, with (3) indicating unchanged. The survey was expected to take approximately 10 min to complete. The survey questions were largely informed by the study by Mel et al. [12].

### 2.3. Data and Statistical Analysis

Using the estimated college student population in the US around the time of COVID-19 inception (19,000,000), with a 95% confidence interval and 5% margin of error, a minimal sample size of 385 would be necessary (SurveyMonkey sample-size calculator). Data were cleaned prior to analysis, ensuring open-text question answers, such as institution, were standardized and accurate for binning of responses. Surveys that had less than 97% completion were removed from the data to ensure completeness of individual data. In the case of small numbers within a category, some data categories were combined prior to analysis. Data were analyzed using the open-source software Jamovi [23,24,25,26,27,28].

Descriptive statistics were assessed using mean and standard deviation (SD) for continuous variables (e.g., age), whereas count (n) and absolute frequencies were used for categorical (e.g., gender) and ordinal (e.g., class year) variables. Given the non-parametric nature of the data, changes in self-reported physical activity before and during COVID-19 were assessed using the Wilcoxon signed-rank test. A Kruskal−Wallis one-way analysis of variance (ANOVA) and/or chi-square analysis for non-parametric data were used to examine differences in physical activity, stress, anxiety, depression, eating and sleeping habits differed by gender, ethnicity, permanent residence in the northeast region, and roommate status. Such data are presented as cross-tabulations. The magnitude of differences between groups were expressed as standardized effect sizes (ES). The rank biserial correlation coefficient was used to determine ES for the Wilcoxon signed-ranked test with threshold values set at small (0.1–0.3), medium (0.3–0.5), and large (>0.5) [29]. The ES threshold values for epsilon squared (ɛ^2^) for Kruskall−Wallis were negligible (0.00 < 0.01), weak (0.01 < 0.04), moderate (0.04 < 0.16), relatively strong (0.16 < 0.36), strong (0.36 < 0.64), and very strong (0.64 < 1.00) [30]. A *p*-value of <0.05 was considered significant.

## 3. Results

### 3.1. Respondents and Demographics

In this study, 655 participants responded to the survey, but only 543 of the responses were valid to use in the experiment, as 112 students were excluded (declined to agree to complete the survey upon reading the consent page, started but did not finish survey, did not agree to have their data used, or was not an undergraduate student). The majority of the respondents were Non-Hispanic White (76%) and female (71.1%), with a mean age of 20.1 (±1.4) years. Most of the respondents attended Skidmore College (42.9%) and reported a permanent residence in the northeast region of the United States (76.8%). Table 1 presents the descriptive analysis of the demographic variables of the participants.

### 3.2. Impact of COVID-19 Restrictions on Exercise Habitus

Based on the Wilcoxon signed-rank test, students reported significantly less days of weekly exercise (*p* = 0.001, ES = 0.663) and a lower intensity of individual exercise sessions (*p* = 0.001, ES = 0.464). Table 2 displays self-reported changes in exercise modality during the COVID-19 restrictions. The most frequent responses for changes in exercise type were “Unaffected by restrictions” (18.3%), “More aerobic training” (17.2%), and “Less aerobic and resistance training” (17.0%). However, as seen in Table 3, individuals that reported a change in their physical health also reported a significant change in their mode of exercise (i.e., aerobic and resistance training) (x^2^ = 245, *p* <0.001). Thus, COVID-19 restrictions significantly influenced the frequency and type of exercise.

### 3.3. Impact of COVID-19 Restrictions on Mental Health and Health Behaviors

As seen in Table 4, there was no significant main effect of gender, ethnicity, permanent residence in the northeast, or roommate status on changes in physical activity level. Significant main effects were found for ethnicity (x^2^ = 17.8, *p* = 0.003, ɛ^2^ = 0.033) and northeast region residence (x^2^ = 7.96, *p* = 0.005, ɛ^2^ = 0.014) on changes in stress (Table 5). The post-hoc Dwass−Steel−Critchlow−Fligner pairwise comparison showed that multi-racial students experienced a significantly higher level of stress compared with Black (*p* = 0.02) and Asian (*p* = 0.04) students during the COVID-19 restrictions. Significant main effects were observed for gender (x^2^ = 16.9, *p* = 0.001, ɛ^2^ = 0.031), ethnicity (x^2^ = 15.9, *p* = 0.007, ɛ^2^ =0.029), and northeast region residence (x^2^ = 6.86, *p* = 0.009, ɛ^2^ =0.012) regarding changes in anxiety (Table 6). The post hoc analysis revealed that males had significantly less anxiety levels compared with females (*p* = 0.01) and non-binary (*p* = 0.017), as well as for Black students compared with multi-racial students (*p* = 0.04) during the COVID-19 restrictions. Significant main effects were found for gender (x^2^ = 16.0, *p* = 0.001, ɛ^2^ =0.029) and ethnicity (x^2^ = 14.9, *p* = 0.011, ɛ^2^ =0.027) regarding changes in depression (Table 7). Non-binary students reported significantly greater levels of depression compared with their male (*p* = 0.001) and female (*p* = 0.007) counterparts. Asian students reported significantly lower levels of depression compared with multi-racial students (*p* = 0.02). No significant main effects were observed for changes in reported eating or sleep behaviors (Table 8 and Table 9). Further, age or class year had no impact on any of the reported outcomes and thus were not presented here. Thus, the COVID-19 restrictions had a major impact on mental health, and impacted region, ethnicity, and gender in disparate ways.

## 4. Discussion

The primary aim of this study was to determine if college students experienced changes in self-reported exercise habits or in anxiety, stress, and depression during COVID-19-related restrictions. Additionally, we sought to examine the effect of independent factors such as living situation and demographics on self-reported exercise habits, anxiety, stress, and depression in response to COVID-19-related restrictions. Respondents reported significant decreases in the number of days per week, as well as the intensity of exercise, during the COVID-19 restrictions; however, they reported no change in minutes of exercise per day, suggesting a shift in their weekly exercise patterns and intensity of each session. Furthermore, there was an apparent shift in the students’ reported mode of exercise, where decreases in resistance exercise were coupled with increases in aerobic exercise, perhaps as a compensatory response to be able to maintain exercise, even if in a different mode. Additionally, students reported increases in anxiety, stress, and depression; we specifically observed a significantly greater negative impact in minority students. These findings suggest that the COVID-19 restrictions had a significant impact on the amount and type of activity students engaged in, and that the impacts of the restrictions were not uniform across student demographics, in that some of the already vulnerable students may have experienced an even greater impact. Administrators and officials should bear this in mind when navigating public-health challenges in the higher education setting.

### 4.1. The Impact of COVID-19 Restrictions on Exercise

Physical activity (PA) and exercise can have profound impacts on student health and wellness. We showed that reduced frequency and intensity of exercise in college students, with varied access to college/university facilities, was in alignment with previous literature [9,10]. Interestingly, though, we did observe that students, somewhat independent of their level of exercise, reported a shift in the mode of exercise they were engaging in, which allowed some students to maintain their level of exercise during the COVID-19 restrictions. Although some researchers reported increases in PA [31], larger scale or trans-institutional studies revealed decreased PA [32,33,34]. The mechanisms responsible for the decreases in exercise reported in the current study, as well the decreased PA in those included in a systematic review [32], may not be able to be determined, but the adoption of adverse health-related behaviors, such as alcohol consumption, may have increased, thus possibly decreasing exercise and PA [35,36]. A strength of the present study was understanding the potential type of activity students were completing, assuming institutional or self-imposed restricted access to fitness centers. Specifically, the predominant shift in exercise training modality we observed was an increased reported amount of aerobic training coupled with a decrease in the reported amount of resistance training students were completing. The potential reasons for this shift could be due to the following: (1) aerobic training can be performed outdoors and is not limited to fitness centers, (2) fitness centers may have been restricted or closed, (3) a desire to reduce potential exposure to the COVID-19 virus with entering a facility, (4) the public-health messaging on cardiovascular exercise possibly reducing the severity of symptoms, and/or (5) an increased focus on one’s lifestyle habits and PA behaviors for overall health. This mode shift was not unique to college students, as it was also observed in the general population, with a shift towards household and recreational PA and away from transportation, occupational, and public-related transit use [13]. Interestingly, and perhaps positively speaking, unlike other reports, we did not observe differences in self-reported physical changes with the COVID-19 restrictions by gender, race/ethnicity, or other sociodemographic factors [33,35]. Such a discrepancy in findings may be due to regional or cultural differences in academic institutions in the United States versus abroad.

Although perhaps unique to the current COVID-19 pandemic, higher education administrators and public-health officials should be aware of such impacts on physical activity and perhaps leverage them in other situations. As colleges/universities continue to battle for student enrollment by offering amenities such as on-campus recreation and fitness facilities, special attention should also be paid to developing or improving outdoor facilities (e.g., walking trails, frisbee golf, and tennis courts) as supplements to indoor fitness training facilities. Further, we do not yet know the long-term implications of these alterations in activity patterns, and future studies should be dedicated to understanding if such alterations have become habit or if they have reverted to pre-pandemic levels in a post-vaccine world.

### 4.2. Impact of the COVID-19 Restrictions on Self-Reported Mental and Physical Health

Students are increasingly experiencing challenges with mental health, such as increased levels of anxiety and/or depression, even pre-pandemic. Our respondents reported a significant increase in stress, anxiety, and depression, with significant differences between genders, ethnicity, and those with permanent residence outside the northeast. This agrees with previous research by Son et al., who found that college students reported a 71% increase in stress and anxiety, as well as with Czeisler et al., who reported a 63% increase in depression and anxiety levels [19,20]. Critically, while Son et al. [19] did not assess the potential influence of student demographic characteristics in mediating the effects of the COVID-19 restrictions, we and Czeisler [20] both reported that such impacts of the COVID-19 restrictions on mental health (depression and anxiety) disproportionately affected students of color. While we cannot ascertain the potential underlying factors or mechanisms driving this disparity, it may relate to perceived or real social/emotional support from family, friends, or the institution (perhaps relating to socioeconomic status) being of minority status among the student population, or friends or family members contracting COVID-19 and developing severe disease or dying. Given the reported racial or ethnic disparities in risk of COVID-19, and subsequent hospitalization or death), the elevated risk among minority students may have weighed on their psyche or may have had loved ones directly impacted by the disease.

It is possible these increases in self-reported levels of anxiety and depression were due to sustaining concerns of contracting COVID-19, academic performance, and/or lifestyle changes [37]. Many students were not acclimated to being removed from friends or family and may have lacked appropriate social support to cope with the increased mental strain. It is important for institutions to be aware of the student population’s mental health, and how the impacts of such large societal changes may influence students in a disparate manner, likely dependent on multiple demographic factors (e.g., gender and race/ethnicity). Specifically, higher education institutions may wish to engage in targeted outreaches, perhaps through various student groups/clubs/organizations, implementing or encouraging students to utilize mental-health or support resources on campus. Additionally, it is recommended that institutions consider providing nature/outdoor-based activities to improve mental-health outcomes and communication regarding mental health with counselors on campus [38].

### 4.3. The Impact of COVID-19 Restrictions on Health-Related Behaviors

Health-related behaviors, such as sleep and diet, above and beyond exercise, can influence student mental and physical health, and may be linked to academic performance. Contrary to what might have been expected, we observed no changes in eating or sleeping habits during the COVID-19 restrictions. This was in contrast with previous findings, which reported improved or compromised eating habits [39,40,41]. As students are heavily reliant on college/university dinning services for their meals, they have limited control over their food choices and are provided relatively high-quality food on a regular basis; some institutions had pick-up or delivery options during the lockdown restrictions. Additionally, as students had already invested financially in a meal plan, they were incentivized to not spend additional resources on snacking or additional food consumption outside of the dining hall. Conversely, students may have moved back home, and with greater time available, may have invested more in dietary planning and/or meal preparation, or their meal choices might have been dictated by their family. However, we did not assess whether students were living on campus or at home, and how the pandemic may have altered living situations in one way or another; this could explain the discrepancies between our study and previous studies and could be the focus of future investigations.

Contrary to previous findings that reported that the majority of students had disruptions in sleep patterns [19,37], we did not observe any change in self-reported sleeping habits. This was surprising as previous research found that increased depression and anxiety were correlated with sleep disruptions [42]. This difference may be due to the timing of data collection, as students may have adjusted to online teaching formats, performed either synchronously (live at scheduled time) or asynchronously (recorded lectures viewable at each student’s choice of time), allowing them to maintain or return to normal sleeping patterns. Alternatively, more objective measures such as actigraphy could have provided a more nuanced look into the potential effects of COVID-19 and associated policies on sleep and its metrics, such as duration, latency, and/or regularity.

### 4.4. Experimental Considerations

There are some limitations to our study that should be mentioned. Our data is representative of small schools primarily from the northeast region of the United States and may not be generalized to larger colleges and universities across the country. Our measures were not objectively collected (e.g., accelerometry-based measures of PA or clinician-based determinations of mental-health status) but were self-reported, and thus, while valuable, should be interpreted with caution and focus on the change in the various outcomes. While there is value in self-reported data, it might only capture part of actual behavior changes. Future investigations should aim to incorporate objective measurements (e.g., accelerometers and dietary recalls), along with subjective assessments. Measures of mental health did not use validated questionnaires; thus, the responses may not have fully captured the intricacies of mental health. Test−retest reliability and validation of such surveys, while not performed in the current study due to the many restrictions of study, may be beneficial for future studies, as well as including more objective measures when possible (e.g., actigraphy of sleep or 24 dietary recall using the ASA24).

## 5. Conclusions

The findings from this study highlight the effects of campus restrictions in response to COVID-19 on exercise and mental health, specifically those most impacted by the restrictions. We observed that students who maintained their exercise level engaged in a modality shift by increasing aerobic activity. However, the largest group of respondents indicated a decrease in aerobic and resistance training. It is unclear whether student exercise behavior returned to normal upon lifting of the restrictions. Furthermore, we observed increases in stress, anxiety, and depression, with minority students reporting the greatest change. Interestingly, we found no effects of the COVID-19 restrictions on dietary or sleep habits.

Considering the continual arms race between institutions and the pivotal role colleges/universities play in the development of long-term behaviors, special attention should be paid to fostering positive relationships with exercise and mental-health counseling. As not all exercise interests for students are the same, diversifying offerings outside of a traditional recreation center could have a greater impact on the student population. Potential additions could include the establishment of walking or hiking trails, outdoor sport courts (e.g., basketball, volleyball, and pickleball), adventure equipment rentals (e.g., tents and kayaks), and bike share programs.

As mental concerns continue to rise on campuses and around the country, institutions should implement programs that foster the development of interpersonal networks; teach healthy coping skills; and provide low-resistance mental-health support such as the Green Bandana project, peer counseling, mentorship programs, and social events. While these programs might be specifically targeted at those greatest at risk, all students would benefit from additional support programs. Understanding student reactions to physical and mental-health challenges, such as seasonal influenza outbreaks or national crises, is crucial, even beyond major pandemics. While success within the classroom may be the primary focus of institutions of higher learning, compromised physical and mental health can compromise that primary objective and should be taken seriously.

## Figures and Tables

**Table 1 ijerph-20-07163-t001:** Participant demographics.

	Total Sample (%)
Gender	
Male	133 (24.5%)
Female	386 (71.1%)
Prefer not to answer or nonbinary	24 (4.4%)
Race/Ethnicity	
Hispanic/Latino	28 (5.2%)
Non-Hispanic White	409 (76.0%)
Black or African American	23 (4.3%)
Asian	25 (4.6%)
Prefer not to disclose	13 (2.6%)
Multiracial	39 (7.2%)
Age (years)	20.1 ± 1.4
Institution Attended	
College of Saint Benedict/St John’s University	13 (2.4%)
Hamilton College	59 (11%)
Mercy College	5 (0.9%)
Skidmore College	231 (42.9%)
St. Lawrence University	100 (18.6%)
Union College	121 (22.5%)
University of Lynchburg	9 (1.7%)
Permanent Residence in the Northeast	
No	124 (22.8%)
Yes	417 (77.2%)
Class	
Freshmen	123 (22.7%)
Sophomore	136 (25.0%)
Junior	133 (24.5%)
Senior	151 (27.8%)
Living with a roommate	
No	126 (23.2%)
Yes	416 (76.8%)

**Table 2 ijerph-20-07163-t002:** Exercise habits prior to and during the COVID-19 restrictions.

	Value (Score ± SD)	*p*-Value	Effect Size
Days of exercise per week			
Prior to lockdowns	6 (7 ± 2)	0.001 *	0.663 ^##^
During lockdowns	3 (4 ± 2)		
Minutes of exercise per day			
Prior to lockdowns	0–29 (3 ± 2)	0.472	0.041
During lockdowns	0–29 (3 ± 2)		
Intensity of exercise per session			
Prior to lockdowns	High intensity (4 ± 1)	0.001 *	0.464 ^#^
During lockdowns	Moderate intensity (3 ± 1)		

* Significant at *p* < 0.05, ^#^ medium effect size, ^##^ large effect size. Data in parentheses are the score ± SD from the Likert-scale survey (Intensity: 3 = moderate intensity, 4 = high intensity).

**Table 3 ijerph-20-07163-t003:** Frequency of self-reported changes in exercise versus shifts in the mode of exercise with the COVID-19 restrictions.

Mode of Exercise		Decreased Significantly	Decreased Modestly	Unchanged	Increased Modestly	Increased Significantly
	Level of Exercise					
Unaffected by restrictions		1 (<1%)	37 (6.9%)	48 (9.0%)	12 (2.2%)	1 (<1%)
More aerobic training		3 (<1%)	36 (6.7%)	23 (4.3%)	27 (5.0%)	3 (<1%)
Less aerobic training		9 (1.6%)	41 (7.7%)	13 (2.4%)	4 (<1%)	0
More resistance training		2 (<1%)	14 (2.6%)	16 (3.0%)	17 (3.2%)	8 (1.5%)
Less resistance training		5 (<1%)	15 (2.8%)	6 (1.1%)	1 (<1%)	0
More aerobic and resistance training		0	5 (<1%)	6 (1.1%)	21 (3.9%)	12 (2.2%)
More aerobic and less resistance training		2 (<1%)	16 (3.0%)	6 (1.1%)	2 (<1%)	1 (<1%)
Less aerobic and more resistance training		2 (<1%)	16 (3.0%)	4 (<1%)	2 (<3%)	1 (<1%)
Less aerobic and resistance training		19 (3.5%)	61 (11.5%)	9 (1.6%)	2 (<1%)	1 (<1%)

Table Note: In the table above, the information for exercise mode is represented in the first column (going down in row), while the level of activity is represented in the 2nd–6th columns. Respondents were able to select “all that apply”, creating multiple permutations outlined in the table above.

**Table 4 ijerph-20-07163-t004:** Self-reported changes in physical activity level by demographic in response to the COVID-19 restrictions.

	1	2	3	4	5	X^2^	*p*-Value
Gender							
Male (n = 132)	12 (9%)	56 (42%)	33 (25%)	22 (16%)	9 (6%)	12.4	0.054
Female (n = 385)	30 (7%)	175 (45%)	96 (24%)	66 (17%)	18 (4%)		
Non-binary (n = 24)	1 (4%)	14 (58%)	7 (29%)	2 (8%)	0 (0%)		
Total (N = 541)	43	245	136	90	27		
Ethnicity							
Hispanic or Latino (n = 28)	5 (2%)	8 (28%)	5 (17%)	5 (17%)	5 (17%)	1.15	0.563
Non-Hispanic or White (n = 407)	28 (6%)	189 (46%)	109 (26%)	62 (15%)	19 (4%)		
Black or African American (n = 23)	2 (8)	10 (43%)	5 (21%)	5 (21%)	1 (4%)		
Asian (n = 25)	1 (4%)	7 (28%)	7 (28%)	8 (32%)	2 (8%)		
Prefer not to disclose (n = 14)	3 (21%)	7 (50%)	4 (28%)	0 (0%)	0 (0%)		
Multiracial (n = 39)	4 (10%)	21 (53%)	6 (15%)	8 (20%)	0 (0%)		
Total (N = 536)	43	242	136	88	27		
From the Northeast region							
No (n = 123)	9 (7%)	57 (46%)	27 (21%)	26 (21%)	4 (3%)	0.06	0.805
Yes (n = 415)	34 (8%)	187 (45%)	109 (26%)	63 (15%)	22 (5%)		
Total (N = 538)	43	144	136	89	26		
Lived with a roommate							
No (n = 409)	34 (8%)	184 (44%)	102 (24%)	66 (16%)	23 (5%)	0.00	0.928
Yes (n = 131)	9 (6%)	60 (45%)	34 (25%)	24 (18%)	4 (3%)		
Total (N = 540)	43	244	136	90	27		

(1) Decreased significantly. (2) Decreased modestly. (3) Unchanged. (4) Increased modestly. (5) Increased significantly.

**Table 5 ijerph-20-07163-t005:** Self-reported changes in stress by demographic in response to the COVID-19 restrictions.

	1	2	3	4	5	X^2^	*p*-Value
Gender							
Male (n = 132)	1 (<1%)	4 (3%)	18 (13%)	65 (49%)	44 (33%)	4.54	0.103
Female (n = 386)	3 (<1%)	12 (3%)	29 (7%)	178 (46%)	164 (42%)		
Non-binary (n = 24)	1 (4%)	1 (4%)	1 (4%)	11 (45%)	10 (41%)		
Total (N = 542)	5	17	48	254	218		
Ethnicity							
Hispanic or Latino (n = 27)	0 (0%)	1 (3%)	1 (3%)	11 (40%)	14 (51%)	17.8	0.003 *
Non-Hispanic or White (n = 409)	4 (<1%)	12 (2%)	32 (7%)	197 (48%)	164 (40%)		
Black or African American (n = 23)	0 (0%)	1 (4%)	7 (23%)	10 (43%)	5 (21%)		
Asian (n = 25)	1 (4%)	2 (8%)	4 (16%)	12 (48%)	6 (24%)		
Prefer not to disclose (n = 14)	0 (0%)	1 (7%)	1 (7%)	7 (50%)	5 (35%)		
Multiracial (n = 39)	0 (0%)	0 (0%)	3 (7%)	14 (35%)	22 (56%)		
Total (N = 537)	5	17	48	251	216		
From the Northeast region							
No (n = 122)	1 (<1%)	6 (4%)	17 (13%)	60 (49%)	38 (31%)	7.96	0.005 *
Yes (n = 417)	4 (<1%)	11 (2%)	31 (7%)	192 (46%)	179 (43%)		
Total (N = 539)	5	17	48	252	217		
Lived with a roommate							
No (n = 131)	1 (<1%)	6 (4%)	13 (9%)	60 (45%)	51 (38%)	0.43	0.512
Yes (n = 410)	4 (<1%)	11 (2%)	35 (8%)	193 (47%)	167 (40%)		
Total (N = 541)	5	18	48	253	218		

(1) Decreased significantly. (2) Decreased modestly. (3) Unchanged. (4) Increased modestly. (5) Increased significantly. * *p* < 0.05.

**Table 6 ijerph-20-07163-t006:** Self-reported changes in anxiety by demographic in response to the COVID-19 restrictions.

	1	2	3	4	5	X^2^	*p*-Value
Gender							
Male (n = 132)	1 (<1%)	3 (2%)	28 (21%)	61 (46%)	39 (29%)	16.9	0.001 *
Female (n = 386)	3 (<1%)	11 (2%)	37 (9%)	153 (39%)	182 (47%)		
Non-binary (n = 24)	0 (0%)	0 (0%)	0 (0%)	12 (50%)	12 (50%)		
Total (N = 542)	4	14	65	226	233		
Ethnicity							
Hispanic or Latino (n = 27)	0 (0%)	1 (3%)	2 (7%)	12 (44%)	12 (44%)	15.9	0.007 *
Non-Hispanic or White (n = 409)	2 (<1%)	10 (2%)	45 (11%)	172 (43%)	180 (44%)		
Black or African American (n = 23)	0 (0%)	2 (8%)	5 (21%)	11 (47%)	5 (22%)		
Asian (n = 25)	1 (4%)	1 (4%)	5 (5%)	12 (48%)	6 (24%)		
Prefer not to disclose (n = 14)	1 (7%)	0 (0%)	4 (28%)	4 (28%)	5 (35%)		
Multiracial (n = 39)	0 (0%)	0 (0%)	4 (10%)	13 (33%)	22 (56%)		
Total (N = 537)	4	14	65	224	230		
From the Northeast region							
No (n = 122)	0 (0%)	5 (4%)	21 (17%)	55 (45%)	41 (33%)	6.86	0.009 *
Yes (n = 417)	4 (<1%)	9 (2%)	44 (10%)	170 (40%)	190 (45%)		
Total (N = 539)	4	14	65	225	231		
Lived with a roommate							
No (n = 131)	1 (<1%)	5 (3%)	16 (12%)	47 (35%)	62 (47%)	0.42	0.513
Yes (n = 410)	3 (<1%)	9 (2%)	49 (11%)	147 (35%)	171 (41%)		
Total (N = 541)	4	14	65	194	233		

(1) Decreased significantly. (2) Decreased modestly. (3) Unchanged. (4) Increased modestly. (5) Increased significantly. * *p* < 0.05.

**Table 7 ijerph-20-07163-t007:** Self-reported changes in depression by demographic in response to the COVID-19 restrictions.

	1	2	3	4	5	X^2^	*p*-Value
Gender							
Male (n = 132)	3 (2%)	3 (2%)	40 (30%)	52 (39%)	34 (25%)	16.0	0.001 *
Female (n = 386)	4 (1%)	12 (3%)	89 (23%)	158 (40%)	131 (33%)		
Non-binary (n = 24)	0 (0%)	1 (6%)	1 (6%)	6 (37%)	16 (66%)		
Total (N = 542)	7	16	130	216	181		
Ethnicity							
Hispanic or Latino (n = 27)	0 (0%)	2 (7%)	7 (25%)	10 (37%)	8 (29%)	14.9	0.011 *
Non-Hispanic or White (n = 409)	5 (1%)	11 (2%)	89 (21%)	160 (39%)	144 (35%)		
Black or African American (n = 23)	1 (4%)	2 (8%)	7 (25%)	9 (33%)	4 (17%)		
Asian (n = 25)	1 (4%)	1 (4%)	9 (36%)	10 (4%)	3 (12%)		
Prefer not to disclose (n = 14)	0 (0%)	0 (0%)	4 (28%)	4 (28%)	6 (42%)		
Multiracial (n = 39)	0 (0%)	0 (0%)	5 (12%)	19 (48%)	15 (38%)		
Total (N = 537)	7	16	121	212	180		
From the Northeast region							
No (n = 122)	1 (<1%)	6 (4%)	31 (25%)	46 (37%)	38 (31%)	1.27	0.259
Yes (n = 417)	6 (1%)	10 (2%)	89 (21%)	189 (45%)	142 (34%)		
Total (N = 539)	7	16	120	235	180		
Lived with a roommate							
No (n = 131)	1 (<1%)	6 (4%)	3 (2%)	60 (45%)	51 (38%)	1.01	0.315
Yes (n = 410)	4 (1%)	11 (2%)	35 (8%)	193 (46%)	167 (40%)		
Total (N = 541)	5	17	38	253	218		

(1) Decreased significantly. (2) Decreased modestly. (3) Unchanged. (4) Increased modestly. (5) Increased significantly. * *p* < 0.05.

**Table 8 ijerph-20-07163-t008:** Self-reported changes in eating habits by demographic in response to the COVID-19 restrictions.

	1	2	3	4	5	X^2^	*p*-Value
Gender							
Male (n = 132)	19 (14%)	52 (39%)	45 (34%)	14 (10%)	2 (1%)	5.22	0.073
Female (n = 386)	59 (15%)	179 (46%)	108 (27%)	40 (10%)	10 (2%)		
Non-binary (n = 24)	6 (25%)	13 (54%)	3 (12%)	2 (8%)	0 (0%)		
Total (N = 542)	84	244	153	56	12		
Ethnicity							
Hispanic or Latino (n = 27)	3 (11%)	14 (51%)	7 (25%)	2 (7%)	1 (3%)	6.86	0.232
Non-Hispanic or White (n = 409)	61 (14%)	193 (47%)	114 (27%)	35 (8%)	6 (1%)		
Black or African American (n = 23)	4 (17%)	8 (34%)	6 (26%)	2 (8%)	3 (13%)		
Asian (n = 25)	1 (4%)	9 (36%)	11 (44%)	3 (12%)	1 (4%)		
Prefer not to disclose (n = 14)	3 (21%)	5 (35%)	6 (42%)	0 (0%)	0 (0%)		
Multiracial (n = 39)	10 (25%)	14 (35%)	11 (28%)	3 (7%)	1 (2%)		
Total (N = 537)	82	243	155	45	12		
From the Northeast region							
No (n = 122)	17 (13%)	53 (43%)	36 (29%)	14 (11%)	2 (1%)	0.93	0.315
Yes (n = 417)	67 (16%)	189 (45%)	120 (28%)	31 (7%)	10 (2%)		
Total (N = 539)	84	242	156	45	12		
Lived with a roommate							
No (n = 131)	23 (17%)	62 (47%)	33 (25%)	9 (6%)	4 (3%)	1.23	0.267
Yes (n = 410)	61 (14%)	182 (44%)	122 (29%)	37 (9%)	8 (1%)		
Total (N = 541)	84	244	155	46	12		

(1) Decreased significantly. (2) Decreased modestly. (3) Unchanged. (4) Increased modestly. (5) Increased significantly.

**Table 9 ijerph-20-07163-t009:** Self-reported changes in sleeping habits by demographic in response to the COVID-19 restrictions.

	1	2	3	4	5	X^2^	*p*-Value
Gender							
Male (n = 132)	21 (15%)	36 (27%)	58 (43%)	13 (9%)	4 (3%)	3.69	0.158
Female (n = 386)	60 (15%)	155 (40%)	119 (30%)	44 (11%)	8 (2%)		
Non-binary (n = 24)	3 (12%)	13 (54%)	7 (29%)	1 (4%)	0 (0%)		
Total (N = 542)	84	191	184	58	12		
Ethnicity							
Hispanic or Latino (n = 27)	7 (25%)	10 (37%)	7 (25%)	3 (11%)	0 (0%)	5.28	0.382
Non-Hispanic or White (n = 409)	56 (13%)	157 (38%)	138 (33%)	50 (12%)	8 (1%)		
Black or African American (n = 23)	5 (21%)	9 (39%)	6 (26%)	2 (8%)	1 (4%)		
Asian (n = 25)	4 (16%)	7 (28%)	11 (44%)	2 (8%)	1 (4%)		
Prefer not to disclose (n = 14)	3 (21%)	5 (35%)	6 (42%)	0 (0%)	0 (0%)		
Multiracial (n = 39)	8 (20%)	15 (38%)	14 (35%)	1 (2%)	1 (2%)		
Total (N = 537)	83	203	182	58	11		
From the Northeast region							
No (n = 122)	19 (15%)	43 (35%)	44 (36%)	15 (12%)	1 (<1%)	0.10	0.751
Yes (n = 417)	64 (15%)	160 (38%)	139 (33%)	43 (10%)	11 (2%)		
Total (N = 539)	83	203	183	58	12		
Lived with a roommate							
No (n = 131)	29 (22%)	46 (35%)	44 (33%)	9 (6%)	3 (2%)	3.65	0.056
Yes (n = 410)	55 (13%)	158 (38%)	140 (34%)	49 (11%)	8 (1%)		
Total (N = 541)	84	204	184	58	11		

(1) Decreased significantly. (2) Decreased modestly. (3) Unchanged. (4) Increased modestly. (5) Increased significantly.

## Data Availability

The data presented in this study may be available on request from the corresponding author.

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
