# Peer review of "The Impact of COVID-19 Health and Safety Measures on the Self-Reported Exercise Behaviors and Mental Health of College Students"

_ijerph, 2023, doi:10.3390/ijerph20247163_

Round 1
Reviewer 1 Report (Previous Reviewer 3)
Comments and Suggestions for Authors
Dear Authors,
I appreciate your submission of the manuscript “The Impact of COVID-19 Health and Safety Measures on the Physical and Mental Health of College Students”. Your research addresses an important topic in the field and contributes valuable insights.
After thoroughly reviewing your manuscript, I offer suggestions for improvement to enhance your work’s overall quality, clarity, and impact. Please remember that these are recommendations, and you can decide how best to address them based on your understanding of the subject matter and your intended audience. Here are the recommendations:
The abstract provides a comprehensive overview of the study and its findings. However, some areas could be improved or clarified for better understanding and coherence. Here are some suggestions:
- Title Reference: The article’s title is not mentioned in the provided section, but ensure it's concise and directly related to the content of the abstract.
- Clarify Restrictions: Instead of just mentioning "public health restrictions", specify what these restrictions entailed briefly, e.g., "remote learning", "campus lockdown", "restricted social gatherings", etc.
- Improve Sentence Flow: Some sentences can be condensed for brevity and clarity. For example:
- "The purpose of this study was to determine if students reduced the number of days, intensity, and minutes of exercise during restrictions. Additionally, if students experienced increased levels of anxiety, stress, and depression during restrictions."
- This could be shortened to: "This study aimed to assess the impact of restrictions on students' exercise habits and their levels of anxiety, stress, and depression."
- Clarify Findings: The abstract mentions both an "increase in aerobic training" and a "combination of reduced aerobic and resistance training". This seems contradictory. Clarify or provide context to this finding.
- Demographics and Regions: Instead of listing each demographic factor, consider summarizing the significant findings: "Demographic factors like gender, ethnicity, and regional residence were found to have significant effects on changes in stress, anxiety, and depression."
- Conclusion: The conclusion can be more precise in emphasizing the importance of providing opportunities for students to exercise. Instead of: "Even in the face of pandemics...", consider: "Amidst pandemics, colleges must prioritize creating opportunities for students to exercise, helping them meet physical activity recommendations and combat mental health issues."
- Language Consistency: The phrase "physical activity and exercise" is repeated. If they're meant to represent different concepts, clarify. Otherwise, consider consistent usage.
- Numerical Data: While abstracts often provide a concise summary, it might be helpful to include specific percentages or numbers (if impactful) for increased anxiety, stress, depression, and changes in exercise habits.
The introduction is comprehensive and provides good background information. Here are some suggested improvements to enhance clarity, coherence, and readability:
- Refine the Opening: The initial description of the virus can be streamlined. Instead of "As a result of the rapid and global spread of the severe acute respiratory system coronavirus 2 (SARS-CoV-2 virus)..." consider "Due to the rapid global spread of the SARS-CoV-2 virus, which can cause symptoms ranging from asymptomatic to severe..."
- Combine Similar Ideas: The impact of COVID-19 restrictions on physical activity and mental health is mentioned multiple times. This can be condensed to avoid redundancy.
- Improve Sentence Flow: Some long sentences can be simplified or divided for better readability. For instance:
- "Regular engagement in PA and exercise has several physical, mental, and social benefits. Currently, the most pertinent are improvements in cardiovascular health..."
- Change to: "Regular engagement in PA and exercise offers numerous benefits, including improved cardiovascular health..."
- Clarify Contradictory Observations: The section discussing how some surveys reported reduced PA in students, while others said increased PA in the general population can be more evident. Consider using contrast words or more explicit transitions.
- Improve Transitions: Ensure smoother transitions between ideas. For example, after discussing the physical activity changes, introduce the new topic of mental health more smoothly. E.g., "While physical activity saw notable changes, the impact on mental health was equally significant."
- Consistency in Terminology: Ensure consistency in the terminology used. This should be clarified if “physical activity” and “exercise” are used interchangeably.
- Highlight the Gap: Make the identified research gap more explicit. Consider: "Despite the documented effects of the pandemic on the general population, there's limited research focusing specifically on college students' physical activity patterns during COVID-19."
- Reframe the Conclusion: The purpose of the study should stand out more. Instead of burying it in the middle of a paragraph, consider starting a new section with "The primary aim of this study is..." or "This study primarily seeks to..."
- Grammar and Word Choice:
- Change "improves in cardiovascular health" to "improvements in cardiovascular health."
- Avoid passive voice where unnecessary, for clarity and brevity.
- Enhance Recruitment Description: Describe the recruitment process in a more structured manner. Follow a linear timeline instead of presenting the recruitment method and consent process.
- Survey Details: Offer more insight into the survey structure.
- Highlight the main sections of the survey, such as demographics, living conditions, and health behaviors.
- Specify whether any validated instruments or scales were used in the study.
- Clarify Data Collection: Improve clarity regarding the information about the policies implemented by institutions. A more concise representation will be helpful.
- Specify Data Cleaning Steps: The data cleaning process can be more detailed.
- Specify what “standardizing” means for open-text questions.
- Clarify the rationale behind the 97% completion cut-off.
- Streamline Statistical Methods: The statistical methods section can be dense for some readers.
- Arrange statistical methods in the order they're used.
- Bullet points or sub-sections could help break down the various methods used.
- Clarity in Categorizing Data: Determine which variables were considered continuous, categorical, or ordinal. This can help readers understand the subsequent analyses.
- Avoid Jargon: Some terms and abbreviations might confuse those unfamiliar with statistical analysis. Where possible, simplify or provide brief explanations.
- Effect Size Ranges: The presentation of effect size (ES) threshold values could be clearer. Consider reformatting for clarity, with tables or more precise delineation.
- Proofread for Typographical Errors: Some errors need correction. For instance:
- The range "moderate (0.04<0.016)" seems to be a typographical error.
- Inconsistent use of the word “and/or”. Choose one as appropriate.
- Clarity in Reporting Data:
- Start with a concise summary of how many participants responded, followed by the number of valid responses, and then the reasons for exclusion. This provides a clear flow.
- Instead of breaking down numbers within the sentence, use bullet points to clarify reasons for exclusion.
- Post-hoc Analysis:
- Specify which group comparisons are being made. I suggest using a list format for easier readability.
- Clear Conclusions:
- Briefly summarise the main findings after each sub-section (e.g., Exercise Habitus, Mental Health). This helps in emphasizing the main takeaways.
- Language and Terminology:
- Ensure that terms like "non-binary", "Black", "Asian" are consistently capitalized and used appropriately.
- Additional Analysis Considerations:
- Mention whether any control factors or adjustments were made when assessing significance (e.g., controlling for socioeconomic status when looking at mental health outcomes).
Here are some suggestions for improvements in the discussion section:
General Points:
- Clarity and Structure: The information is dense, and the section can benefit from clearer structure and subheadings.
- Transition Phrases: Use transition phrases to enhance flow and readability.
- Diversification of Sources: Although various sources are cited, diversifying with more recent or global studies might provide a richer context.
4. Discussion:
- Begin with a summary of the main findings before diving into specifics.
- Emphasize the main purpose and methodology in the initial section to reiterate the context.
4.1. The Impact of COVID-19 Restrictions on Physical Activity
- Instead of jumping directly into the findings, first highlight the importance of physical activity in the context of student health and wellness.
- Clarify terms like "PA" upon first mention.
- Draw parallels to pre-pandemic exercise patterns, indicating the broader implications of these changes on overall student health.
4.2. The Impact of COVID-19 Restrictions on Self-Reported Mental and Physical Health
- Early on, provide more context on the importance of mental health in student populations.
- Emphasize that the demographic-specific impacts may require unique interventions.
- Given the stark findings, discuss potential preventative measures colleges could implement to aid mental health.
4.3. The Impact of COVID-19 Restrictions on Health-Related Behaviors
- Clarify the link between eating and sleeping habits and mental health.
- Incorporate information on the importance of sleep for college students, given its relevance to academic performance, mental health, and general well-being.
- Given the contrast between this study's findings and previous ones, elaborate on potential reasons for these discrepancies.
4.4. Experimental Considerations
- Highlight that while self-reported data provides valuable insights into perceived changes, it might only capture part of actual behavior changes.
- Discuss potential biases in self-reported data.
- Emphasize the need for further research using objective measures for a more comprehensive understanding.
Additional Points:
- Inclusion of Potential Interventions: Given the stark findings, especially mental health-related, provide more specific recommendations for institutions.
- Emphasis on Disparities: While the text touches on disparities in impacts based on race and ethnicity, there might be a need to delve deeper into the implications of these findings and how institutions can address them.
While the conclusion does a good job summarizing the key findings of the study and suggesting some potential interventions for colleges and universities, there are a few areas that could be improved or expanded upon:
- Clarity and Focus: The conclusion seems to jump from one topic to another. It might be more coherent to start with a clear summarization of the main findings and then delve into implications and recommendations.
- Generalizability Warning: Given the limitations stated earlier in the article, the conclusions could reiterate the caution that these findings primarily apply to small schools in the northeast and may not be generalizable to larger institutions or different regions.
- Broader Implications: While focusing on college students is essential, it might be helpful to discuss the wider implications of these findings briefly. How do these results fit a more extensive picture regarding the impact of lockdowns or similar restrictions on different populations?
- Future Research: While the study did find no effects on dietary or sleep habits, the conclusion could acknowledge that these are complex behaviors influenced by multiple factors. A suggestion for future research could be to further explore these behaviors using a combination of subjective and objective measurements.
- Specific Interventions: While the conclusion does suggest some interventions like trails and outdoor sport courts, it could be more explicit about the kinds of programs that might foster the development of interpersonal networks. For example, mentorship programs, peer counseling, or social events tailored to minority students could be mentioned.
- Addressing Mental Health: Emphasize the importance of developing comprehensive mental health programs on campus, given the rise in reported anxiety, stress, and depression. While focusing on minority students is critical, all students could benefit from increased mental health resources.
- Historical Context: The mention of other pandemics is valuable, but the phrasing "even if seasonal influenza" might seem to minimize the impact of influenza. It might be clearer to say that understanding student reactions to health challenges, such as seasonal influenza outbreaks, is crucial even beyond major pandemics.
- Call to Action: It might be powerful to end with a strong call to action, urging institutions to take these findings seriously and to be proactive in designing programs and infrastructure that prioritize student health, especially in times of crisis.
The paper is generally well-written. However, there are some areas where clarity and fluency could be improved:
Sentence Structure: Some sentences are pretty lengthy,
Consistency in Terminology: Ensure that the same terms are used consistently throughout
Tense Agreement: Ensure the consistent use of tenses. If you're describing findings from the study, use the past tense. The present tense is appropriate if you're discussing general facts or cited research.
Redundancy: Avoid repeating the same idea in different ways within a short span of the text.
Author Response
I appreciate your submission of the manuscript “The Impact of COVID-19 Health and Safety Measures on the Physical and Mental Health of College Students”. Your research addresses an important topic in the field and contributes valuable insights.
After thoroughly reviewing your manuscript, I offer suggestions for improvement to enhance your work’s overall quality, clarity, and impact. Please remember that these are recommendations, and you can decide how best to address them based on your understanding of the subject matter and your intended audience. Here are the recommendations:
RESPONSE: We appreciate the care that the reviewer has put into their suggestions for improving the manuscript. We have taken the overwhelming majority of recommendations into revising the paper. Thank you.
The abstract provides a comprehensive overview of the study and its findings. However, some areas could be improved or clarified for better understanding and coherence. Here are some suggestions:
- Title Reference: The article’s title is not mentioned in the provided section, but ensure it's concise and directly related to the content of the abstract.
RESPONSE: We have modified the title for improved clarity
- Clarify Restrictions: Instead of just mentioning "public health restrictions", specify what these restrictions entailed briefly, e.g., "remote learning", "campus lockdown", "restricted social gatherings", etc.
RESPONSE: We have added examples of implemented restrictions.
- Improve Sentence Flow: Some sentences can be condensed for brevity and clarity. For example:
- "The purpose of this study was to determine if students reduced the number of days, intensity, and minutes of exercise during restrictions. Additionally, if students experienced increased levels of anxiety, stress, and depression during restrictions."
- This could be shortened to: "This study aimed to assess the impact of restrictions on students' exercise habits and their levels of anxiety, stress, and depression."
RESPONSE: We have made this change for improved brevity and clarity.
- Clarify Findings: The abstract mentions both an "increase in aerobic training" and a "combination of reduced aerobic and resistance training". This seems contradictory. Clarify or provide context to this finding.
RESPONSE: We have added more to this statement. We hope this clarifies the findings.
- Demographics and Regions: Instead of listing each demographic factor, consider summarizing the significant findings: "Demographic factors like gender, ethnicity, and regional residence were found to have significant effects on changes in stress, anxiety, and depression."
RESPONSE: We have used the reviewers revised statement for brevity.
- Conclusion: The conclusion can be more precise in emphasizing the importance of providing opportunities for students to exercise. Instead of: "Even in the face of pandemics...", consider: 3"Amidst pandemics, colleges must prioritize creating opportunities for students to exercise, helping them meet physical activity recommendations and combat mental health issues."
RESPONSE: We have modified this sentence for a greater call to action and simplicity.
- Language Consistency: The phrase "physical activity and exercise" is repeated. If they're meant to represent different concepts, clarify. Otherwise, consider consistent usage.
RESPONSE: We have removed the term “physical activity” and are now only using the term exercise in the abstract.
- Numerical Data: While abstracts often provide a concise summary, it might be helpful to include specific percentages or numbers (if impactful) for increased anxiety, stress, depression, and changes in exercise habits.
RESPONSE: This is a great suggestion. We have included the percent change in for minutes of exercise.
The introduction is comprehensive and provides good background information. Here are some suggested improvements to enhance clarity, coherence, and readability:
- Refine the Opening: The initial description of the virus can be streamlined. Instead of "As a result of the rapid and global spread of the severe acute respiratory system coronavirus 2 (SARS-CoV-2 virus)..." consider "Due to the rapid global spread of the SARS-CoV-2 virus, which can cause symptoms ranging from asymptomatic to severe..."
RESPONSE: We have used the reviewer’s recommended revised statement.
- Combine Similar Ideas: The impact of COVID-19 restrictions on physical activity and mental health is mentioned multiple times. This can be condensed to avoid redundancy.
- RESPONSE: We have tried to combine where possible
- Improve Sentence Flow: Some long sentences can be simplified or divided for better readability. For instance:
- "Regular engagement in PA and exercise has several physical, mental, and social benefits. Currently, the most pertinent are improvements in cardiovascular health..."
- Change to: "Regular engagement in PA and exercise offers numerous benefits, including improved cardiovascular health..."
- RESPONSE: We have revised the statement to the reviewer’s recommended version.
- Clarify Contradictory Observations: The section discussing how some surveys reported reduced PA in students, while others said increased PA in the general population can be more evident. Consider using contrast words or more explicit transitions.
- RESPONSE: We made revisions based on this
- Improve Transitions: Ensure smoother transitions between ideas. For example, after discussing the physical activity changes, introduce the new topic of mental health more smoothly. E.g., "While physical activity saw notable changes, the impact on mental health was equally significant."
- RESPONSE: Thank you for this suggestion, this has been edited.
- Consistency in Terminology: Ensure consistency in the terminology used. This should be clarified if “physical activity” and “exercise” are used interchangeably.
- RESPONSE: We have removed the interchange of “physical activity” and “Exercise”.
- Highlight the Gap: Make the identified research gap more explicit. Consider: "Despite the documented effects of the pandemic on the general population, there's limited research focusing specifically on college students' physical activity patterns during COVID-19."
- RESPONSE: We have used the reviewer’s revised version.
- Reframe the Conclusion: The purpose of the study should stand out more. Instead of burying it in the middle of a paragraph, consider starting a new section with "The primary aim of this study is..." or "This study primarily seeks to..."
- RESPONSE: We have made this change by separating out the conclusion and aim of the study.
- Grammar and Word Choice:
- Change "improves in cardiovascular health" to "improvements in cardiovascular health."
RESPONSE: We have made this wording change.
- Avoid passive voice where unnecessary, for clarity and brevity.
- Enhance Recruitment Description: Describe the recruitment process in a more structured manner. Follow a linear timeline instead of presenting the recruitment method and consent process.
- RESPONSE: We have reworded the section to provide a more linear timeline.
- Survey Details: Offer more insight into the survey structure.
- Highlight the main sections of the survey, such as demographics, living conditions, and health behaviors.
- RESPONSE: We have separated out this section to highlight individual survey sections.
- Specify whether any validated instruments or scales were used in the study.
- Highlight the main sections of the survey, such as demographics, living conditions, and health behaviors.
RESPONSE: Our survey and questions were largely based off of a prior study by Mel et al (ref#11), and is now mentioned in the article.
- Clarify Data Collection: Improve clarity regarding the information about the policies implemented by institutions. A more concise representation will be helpful.
RESPONSE: We appreciate this comment, while many of the institutional restrictions were likely similar based on CDC guidance but further refined by local (county/State) health authorities, putting all of the restrictions from the institutions and for the various time periods would be daunting for author and reader alike.
- Specify Data Cleaning Steps: The data cleaning process can be more detailed.
- Specify what “standardizing” means for open-text questions.
- RESPONSE: To allow for multiple institutions this was open text, and if respondents answered, “skidmore”, Skidmore”, “Skidmore college” or “Skidmore College, these were all treated as unique answers, and thus were edited to ensure adequate binning. The same was true for state (e.g. NY vs. New York vs. new york).
- Clarify the rationale behind the 97% completion cut-off.
- RESPONSE: We have clarified this point, the majority met this criterion, which allows for some of the cross comparisons we wanted to make and a 50% completion wouldn’t have allowed for such comparisons.
- Streamline Statistical Methods: The statistical methods section can be dense for some readers.
- Arrange statistical methods in the order they're used.
- Bullet points or sub-sections could help break down the various methods used.
- RESPONSE: We have made some minor revisions here, but the statistical method does align with the order of presentation (Descriptives first, PA changes second, etc).
- Clarity in Categorizing Data: Determine which variables were considered continuous, categorical, or ordinal. This can help readers understand the subsequent analyses.
- RESPONSE: We have added examples to help clarify, thank you for this suggestion.
- Avoid Jargon: Some terms and abbreviations might confuse those unfamiliar with statistical analysis. Where possible, simplify or provide brief explanations.
- RESPONSE: We have made some targeted edits here.
- Effect Size Ranges: The presentation of effect size (ES) threshold values could be clearer. Consider reformatting for clarity, with tables or more precise delineation.
- RESPONSE: We retain the organization to avoid adding additional tables.
- Proofread for Typographical Errors: Some errors need correction. For instance:
- The range "moderate (0.04<0.016)" seems to be a typographical error.
- We have fixed this typographical error.
- Inconsistent use of the word “and/or”. Choose one as appropriate.
- The range "moderate (0.04<0.016)" seems to be a typographical error.
- Clarity in Reporting Data:
- Start with a concise summary of how many participants responded, followed by the number of valid responses, and then the reasons for exclusion. This provides a clear flow.
- Instead of breaking down numbers within the sentence, use bullet points to clarify reasons for exclusion.
- Specify what “standardizing” means for open-text questions.
RESPONSE: We have revised this modestly to improve clarity.
- Post-hoc Analysis:
- Specify which group comparisons are being made. I suggest using a list format for easier readability.
RESPONSE: We appreciate this suggestion, but as no other reviewer made the same suggestion we will stick with the traditional format, though we like the idea.
- Clear Conclusions:
- Briefly summarise the main findings after each sub-section (e.g., Exercise Habitus, Mental Health). This helps in emphasizing the main takeaways.
- RESPONSE: We have included such summaries, thank you.
- Language and Terminology:
- Ensure that terms like "non-binary", "Black", "Asian" are consistently capitalized and used appropriately.
- RESPONSE: We have made this correction.
- Additional Analysis Considerations:
- Mention whether any control factors or adjustments were made when assessing significance (e.g., controlling for socioeconomic status when looking at mental health outcomes).
- RESPONSE: While we appreciate this suggestion we did not measure any indicator of socioeconomic status, and given the focus was on college-students there is a range of socioeconomic statuses, ranging from first generation students of color to legacy white students of affluence. Despite these potential differences, factors like race/ethnicity and gender were significant with regards to mental health impacts.
Here are some suggestions for improvements in the discussion section:
General Points:
- Clarity and Structure: The information is dense, and the section can benefit from clearer structure and subheadings.
- Transition Phrases: Use transition phrases to enhance flow and readability.
- Diversification of Sources: Although various sources are cited, diversifying with more recent or global studies might provide a richer context.
RESPONSE: We appreciate these comments, we have made some edits to the discussion to improve the manuscript.
- Discussion:
- Begin with a summary of the main findings before diving into specifics.
- Emphasize the main purpose and methodology in the initial section to reiterate the context.
- RESPONSE: We have started each section of the discussion by summarizing the findings.
4.1. The Impact of COVID-19 Restrictions on Physical Activity
- Instead of jumping directly into the findings, first highlight the importance of physical activity in the context of student health and wellness.
- Clarify terms like "PA" upon first mention.
- Draw parallels to pre-pandemic exercise patterns, indicating the broader implications of these changes on overall student health.
RESPONSE: Thank you for these suggestions we have revised accordingly.
4.2. The Impact of COVID-19 Restrictions on Self-Reported Mental and Physical Health
- Early on, provide more context on the importance of mental health in student populations.
- Emphasize that the demographic-specific impacts may require unique interventions.
- Given the stark findings, discuss potential preventative measures colleges could implement to aid mental health.
- RESPONSE: Thank you for these suggestions we have revised accordingly.
4.3. The Impact of COVID-19 Restrictions on Health-Related Behaviors
- Clarify the link between eating and sleeping habits and mental health.
- Incorporate information on the importance of sleep for college students, given its relevance to academic performance, mental health, and general well-being.
- Given the contrast between this study's findings and previous ones, elaborate on potential reasons for these discrepancies.
- RESPONSE: Thank you for these suggestions we have revised accordingly.
4.4. Experimental Considerations
- Highlight that while self-reported data provides valuable insights into perceived changes, it might only capture part of actual behavior changes.
- Discuss potential biases in self-reported data.
- Emphasize the need for further research using objective measures for a more comprehensive understanding.
- RESPONSE: Thank you for this suggestion we have revised accordingly.
Additional Points:
- Inclusion of Potential Interventions: Given the stark findings, especially mental health-related, provide more specific recommendations for institutions.
- Emphasis on Disparities: While the text touches on disparities in impacts based on race and ethnicity, there might be a need to delve deeper into the implications of these findings and how institutions can address them.
- RESPONSE: Thank you for this suggestion we have considered these points in our revisions.
While the conclusion does a good job summarizing the key findings of the study and suggesting some potential interventions for colleges and universities, there are a few areas that could be improved or expanded upon:
- Clarity and Focus: The conclusion seems to jump from one topic to another. It might be more coherent to start with a clear summarization of the main findings and then delve into implications and recommendations.
- RESPONSE: Thank you for this suggestion we have refined these sections for improved clarity.
- Generalizability Warning: Given the limitations stated earlier in the article, the conclusions could reiterate the caution that these findings primarily apply to small schools in the northeast and may not be generalizable to larger institutions or different regions.
- Broader Implications: While focusing on college students is essential, it might be helpful to discuss the wider implications of these findings briefly. How do these results fit a more extensive picture regarding the impact of lockdowns or similar restrictions on different populations?
- Future Research: While the study did find no effects on dietary or sleep habits, the conclusion could acknowledge that these are complex behaviors influenced by multiple factors. A suggestion for future research could be to further explore these behaviors using a combination of subjective and objective measurements.
- RESPONSE: We appreciate this suggestion. We have added recommendations for future research.
- Specific Interventions: While the conclusion does suggest some interventions like trails and outdoor sport courts, it could be more explicit about the kinds of programs that might foster the development of interpersonal networks. For example, mentorship programs, peer counseling, or social events tailored to minority students could be mentioned.
- RESPONSE: Thank you for this suggestion, we have added examples current interventions and suggestions for new programs.
- Addressing Mental Health: Emphasize the importance of developing comprehensive mental health programs on campus, given the rise in reported anxiety, stress, and depression. While focusing on minority students is critical, all students could benefit from increased mental health resources.
- RESPONSE: Thank you for this recommendation, we have added content on this topic.
- Historical Context: The mention of other pandemics is valuable, but the phrasing "even if seasonal influenza" might seem to minimize the impact of influenza. It might be clearer to say that understanding student reactions to health challenges, such as seasonal influenza outbreaks, is crucial even beyond major pandemics.
- RESPONSE: Great recommendation, we have refined this statement.
- Call to Action: It might be powerful to end with a strong call to action, urging institutions to take these findings seriously and to be proactive in designing programs and infrastructure that prioritize student health, especially in times of crisis.
RESPONSE: Thank you, we have revised this section, largely based on these suggestions.
Reviewer 2 Report (New Reviewer)
Comments and Suggestions for Authors
1. What is the main question addressed by the research?
This paper aimed to investigate determine if students reduced the number of days, intensity, and minutes of exercise during restrictions due to COVID-19 pandemic. Additionally, if students experienced increased levels of anxiety, stress, and depression during restrictions.
2. Do you consider the topic original or relevant in the field? Does it address a specific gap in the field?
No, the topic is relevant but not original. The COVID-19 era is passed …. There were so many studies during that time regarding PA and mental health.
3. What does it add to the subject area compared with other published material?
As mentioned before, there were so many studies during that time regarding PA and mental health. So, the results of this study are almost similar ….
4. Has the introduction of the research explained the problem well?
Yes, the introduction is well written and the problem for conducting this paper is well constructed.
5. What specific improvements should the authors consider regarding the methodology? What further controls should be considered?
Methodology is properly formulated.
6. Are the results well-structured and presented?
Yes, the results are clear and well presented.
7. Are the discussion and conclusions consistent with the evidence and arguments presented and do they address the main question posed?
- Please add some practical implications for the findings of this paper.
8. Are the references appropriate?
Yes, they are relevant and appropriate.
The paper is well written (except my minor comment above), but due to my first comment that the topic is not original, my decision is “reject”.
Comments on the Quality of English LanguageMinor improvement is required.
Author Response
What is the main question addressed by the research?
This paper aimed to investigate determine if students reduced the number of days, intensity, and minutes of exercise during restrictions due to COVID-19 pandemic. Additionally, if students experienced increased levels of anxiety, stress, and depression during restrictions.
RESPONSE: We appreciate the time and effort of the reviewer in helping to improve the manuscript.
- Do you consider the topic original or relevant in the field? Does it address a specific gap in the field?
No, the topic is relevant but not original. The COVID-19 era is passed …. There were so many studies during that time regarding PA and mental health.
RESPONSE: We appreciate the concern of the reviewer; however, replication is a hallmark of science and this study can provide data for systematic review and meta-analysis which may be able to provide stronger insight into the impacts of COVID-19. Which, while the emergency health declaration has been recanted, studying the impact is still worthy of attention. We might not yet fully appreciate the impacts that the time period had on individuals. For example, in the Dutch famine studies of world war 2, another major global event, effects were seen years later even though the event had passed. While we understand the reviewer’s sentiment, we respectfully disagree that these findings may provide insight for follow up and if we think that the era of having any pandemics is over, we would have made a gross error, so any lessons we can learn for the future are also likely of value.
- What does it add to the subject area compared with other published material?
As mentioned before, there were so many studies during that time regarding PA and mental health. So, the results of this study are almost similar ….
RESPONSE: the reviewer brings up an interesting point, which is that the results are “almost similar” except in a few cases they were not, but in fact contradictory. Thus, while likely context dependent (e.g. country, institution type, etc.), we are still figuring out the impacts.
- Has the introduction of the research explained the problem well?
Yes, the introduction is well written and the problem for conducting this paper is well constructed.
RESPONSE: We appreciate the kind feedback on the paper
- What specific improvements should the authors consider regarding the methodology? What further controls should be considered?
Methodology is properly formulated.
RESPONSE: We appreciate this recognition of the paper
- Are the results well-structured and presented?
Yes, the results are clear and well presented.
RESPONSE: Thank you for this acknowledgement.
- Are the discussion and conclusions consistent with the evidence and arguments presented and do they address the main question posed?
- Please add some practical implications for the findings of this paper.
RESPONSE: We have revised the discussion, especially the conclusion in this regard.
- Are the references appropriate?
Yes, they are relevant and appropriate.
RESPONSE: We appreciate the kind feedback on the paper
The paper is well written (except my minor comment above), but due to my first comment that the topic is not original, my decision is “reject”
RESPONSE: We appreciate the candor of the reviewer, though as stated above, originality while valuable isn’t as important as replication, as far too many fields or topics have fell victim to the replication crisis, where there is little replication and some of the seminal findings have been found to be refuted or inaccurate.
Reviewer 3 Report (New Reviewer)
Comments and Suggestions for Authors
Thank you for reviewing the paper titled “The Impact of COVID-19 Health and Safety Measures on the 2 Physical and Mental Health of College Students.” The paper is organized, well-written, but some question raised into consideration:
Title: I recommend to write type of study at the end of the title.
Introduction: need more example on studies which is similar or against to this one.
Mention validity and reliability for the questionnaire.
“Sample size was calculated” Where is sample size calculation? Also mention the test name.
“institutions beyond the 83 NY6”.To what exactly is the author referring.
112 students declined to agree to complete the survey upon reading the consent page, started but did not finish survey. Mention the reason of discontinuation the questionnaire.
Mention inclusion and exclusion criteria for the study please
Discussion: unfortunately, does not cover all results
Some sentences of conclusion should be better addressed in the discussion section.
Check the manuscript English language and grammar
Check references well
Thank you for reviewing the paper titled “The Impact of COVID-19 Health and Safety Measures on the 2 Physical and Mental Health of College Students.” The paper is organized, well-written, but some question raised into consideration:
Title: I recommend to write type of study at the end of the title.
Introduction: need more example on studies which is similar or against to this one.
Mention validity and reliability for the questionnaire.
“Sample size was calculated” Where is sample size calculation? Also mention the test name.
“institutions beyond the 83 NY6”.To what exactly is the author referring.
112 students declined to agree to complete the survey upon reading the consent page, started but did not finish survey. Mention the reason of discontinuation the questionnaire.
Mention inclusion and exclusion criteria for the study please
Discussion: unfortunately, does not cover all results
Some sentences of conclusion should be better addressed in the discussion section.
Check the manuscript English language and grammar
Check references well
Comments on the Quality of English Language
Check the manuscript English language and grammar
Author Response
Thank you for reviewing the paper titled “The Impact of COVID-19 Health and Safety Measures on the 2 Physical and Mental Health of College Students.” The paper is organized, well-written, but some question raised into consideration:
RESPONSE: We appreciate the effort of the reviewer and for their constructive and kind feedback.
Title: I recommend to write type of study at the end of the title.
RESPONSE: We have edited the title to reflect this
Introduction: need more example on studies which is similar or against to this one.
RESPONSE: We have added additional references in this area to better encompass the state of the literature.
Mention validity and reliability for the questionnaire.
“Sample size was calculated” Where is sample size calculation? Also mention the test name.
RESPONSE: We used the survey monkey sample size estimator, and is now stated in the manuscript.
Lines 105-107
“institutions beyond the 83 NY6”.To what exactly is the author referring.
RESPONSE: The NY6 is a consortium of 6 liberal arts colleges in upstate New York, and is now clarified in the manuscript.
112 students declined to agree to complete the survey upon reading the consent page, started but did not finish survey. Mention the reason for discontinuation of the questionnaire.
RESPONSE: Subjects didn’t list a reason for discontinuation of the survey. As survey was applied remotely through Qualtrics, we were not able to capture that data. Several assumptions can be made as to rational such as; length of survey, survey questions, or forget to complete the survey once started. Frustratingly, after consenting to complete the study, a surprising number didn’t agree to have their data used for the study which was the last page/question in the survey. As they discontinued or failed to complete it is impossible to provide any understanding as to why.
Mention inclusion and exclusion criteria for the study please
RESPONSE: These are now stated in the paper (line 150)
Discussion: unfortunately, does not cover all results
Some sentences of conclusion should be better addressed in the discussion section.
Check the manuscript English language and grammar
Check references well
RESPONSE: We have significantly revised the discussion in response to this and other reviewer comments and believe it is now improved as a result.
Round 2
Reviewer 1 Report (Previous Reviewer 3)
Comments and Suggestions for Authors
Dear Authors,
I want to extend my heartfelt thanks for accepting my comments and considering them for revision in your manuscript. Thank you again, and I look forward to seeing the final version of your paper.
Author Response
Thanks!
Reviewer 2 Report (New Reviewer)
Comments and Suggestions for Authors
Thanks for your reply.
Author Response
Thanks!
This manuscript is a resubmission of an earlier submission. The following is a list of the peer review reports and author responses from that submission.
Round 1
Reviewer 1 Report
Comments and Suggestions for Authors
This article has been discussed about the impact of COVID-19 restrictions on the Physical and Mental Health of College Students. They talked about the changes from the responses by gender, ethnicity, region and with/without roommate which could help to understand the effects to people’s living habits. Here are some suggestions/questions for minor changes:
1. Based on the design of the survey, it is asking some questions on individual demographics, living arrangements, physical activity sleep, diet, and mental health before and after the start of the COVID-19 pandemic. Talking about the ‘before and after the start of the COVID-19 pandemic’, people will have different understandings of the time range, have the survey claimed a clear time period for the pandemic? Since that may affect the description of lifestyle changes for different people
2. For the survey result, it could be better to use some plots to insight the increase/decrease
3. Could you revise this sentence in section 2.1. Participants and General Procedures:
“Participants were primarily recruited from Skidmore College and higher education institutions in the New York 6 (Union College, Hamilton College, Colgate College, St. Lawrence University, Hobart and William Smith Colleges) and beyond.”
Author Response
Reviewer#1
This article has been discussed about the impact of COVID-19 restrictions on the Physical and Mental Health of College Students. They talked about the changes from the responses by gender, ethnicity, region and with/without roommate which could help to understand the effects to people’s living habits. Here are some suggestions/questions for minor changes:
RESPONSE: We would like to thank the reviewer for their insightful and constructive comments, as we believe they have helped strengthen the manuscript. We have responded in a point-by-point fashion below.
- Based on the design of the survey, it is asking some questions on individual demographics, living arrangements, physical activity sleep, diet, and mental health before and after the start of the COVID-19 pandemic. Talking about the ‘before and after the start of the COVID-19 pandemic’, people will have different understandings of the time range, have the survey claimed a clear time period for the pandemic? Since that may affect the description of lifestyle changes for different people.
RESPONSE: The survey was sent out, and open to responses from 2/1/2021 to 3/10/2021, responses ranged from 2/2/2021 to 3/9/2021. This has been clarified in the text.
- For the survey result, it could be better to use some plots to insight the increase/decrease
RESPONSE: We had the same thought as the reviewer. However, we attempting to add figures, we could not find a figure format the quantity of data in a reader friendly format. Additionally, with the current volume of tables we did not feel they significantly improved the manuscript.
- Could you revise this sentence in section 2.1. Participants and General Procedures:
“Participants were primarily recruited from Skidmore College and higher education institutions in the New York 6 (Union College, Hamilton College, Colgate College, St. Lawrence University, Hobart and William Smith Colleges) and beyond.”
RESPONSE: We have revised this sentence, thank you.
Reviewer 2 Report
Comments and Suggestions for Authors
dear authors,
please check the attachment for my comments.
also I would like to know the reason for delay between survey time and reporting the results.
Good luck

Author Response
REVIEWER#2
In this study authors try to assess the state of physical activity and mental health and other health behaviors (eating and sleep) regarding the restrictions of covid-19. Measurements were not objective and were self-reported.
The introduction is very organized and give us a good picture of the problem. The methodology is very well described. Results and discussion section needs to be improved.
RESPONSE: We would like to thank the reviewer for their insightful and constructive comments, as we believe they have helped strengthen the manuscript. We have responded in a point-by-point fashion below.
My comments are as follows
1) Line 32, please recheck the date when WHO declare covid-19 as pandemic. It was on March 11, 2020.
RESPONSE: Thank you this has since been updated accordingly.
2) Line 39-40, a reference is needed for the statement that physical activity improve body response to civid-19 vaccine and decrease the severity of its symptoms.
RESPONSE: Edited, references were misplaced one sentence later.
3) Line 42, impact "on"
RESPONSE: Edited, thank you.
4) Line 46, due "to"
RESPONSE: Edited, thank you.
5) Line 46, "colleges"
RESPONSE: Edited, thank you.
6) Line 68, due "to"
RESPONSE: Edited, thank you.
7) Please provide information on the status of restrictions at the time of this study. Were students still under restrictions at the time of survey or the questions ask them to remember previous time? If so, how do you manage recall bias?
RESPONSE: Thank you for this comment, we have since added further information in the methods (Line 96-100). While specific to one institution (link https://www.skidmore.edu/president/posts/2021/0115-spring-semester-plan-update.php) others likely followed similar isolation guidelines. Additional quarantines were conducted on an as needed exposure basis using contact tracing.
8) Could you please provide the response rate? (response/ view)
RESPONSE: We received 655 responses, but only used 543, as we excluded those who: had a less than 97% completion rate of questions, answered “no” to wanting to submit their data, or anyone who did not press “I agree to the consent statement”.
- 9) Please provide more details on the variables. a. Is "Living in northeast" a separate variable? What were other categories? Why northeast is important?
- Was Intensity gathered as a qualitative variable or a quantitative variable? In the methods section it is referred as a qualitative variable, while in table 2 it is reported as mean and SD.
- Was duration gathered as a qualitative variable or a quantitative variable? It seems to be a quantitative variable, but in table 2, it is reported as a qualitative and quantitative variable.
- Could you provide data on living in dormitory?
- From table 8, it seems that participants have to answer the question on "changes in eating habit". The Likert scale is unclear to me here. For example, what is the meaning of "increased significantly in eating habit"?
- RESPONSE:
- a) “Living in the northeast” was not a direct variable obtained. Instead students were asked for their state of permanent residence, from this information we were able to determine if they are from the Northeast region of the United States. This variable was included to explore the ideas of potential support systems in the region (e.g., friends, family) and/or familiarity/comfort with the area.
- b) Duration was gathered as a qualitative/self-report variable. Respondents were provided 6 different options to describe their exercise duration (e.g., no exercise, 0-29 minutes, 30-59 minutes). The mean and standard deviation values provided is there to provide more detail on the duration of exercise, although the average response for exercise was 0-29 minutes for pre and during lockdowns, we did see that some students were not exercising at all while others were exercising between 60-89 minutes. We do plan to include the survey questions to allow readers additional context.
- c) The same methodology was used for intensity as duration.
- d) We do not specifically have dormitory status to provide, other than if they lived alone or with a roommate.
- e) Thank you for bringing up this piece of clarification. For eating habits, the questions were slightly modified. The question was “How has the pandemic effected your eating habits?”, with responses ranging from “Significantly worse” to “Significantly better”.
And also for sleeping habit, what is the meaning of increase or decrease in sleeping habit?
10) Table 1, please define the numbers. Is it mean and SD or number and percent?
RESPONSE: Responses for sleeping habits were similarly modified, options ranged from “Significantly worse” to “Significantly better”
11) Table 2, please provide all the categories for intensity.
RESPONSE: We will include the entire survey as supplemental material. Thank you
12) Table 3 is not clear. I do not understand the row one. For example what is the meaning of "decrease in unaffected by restrictions"? if a participants physical activity did not changed by restrictions, how could they define that it was decreased or increased?
RESPONSE: We apologize for any confusion, the table is a cross-tabulation table, we have edited the table header, and added a footer to better clarify the table. Briefly, the first column and rows within it, represent the mode of exercise and possible COVID-19 associated changes, whereas the columns represent possible changes in activity, allowing examination of the various permutations of outcomes concomitant with the chi-square analysis that was employed. Thank you we believe this is better clarified now.
13) The percent should be provided in table 3.
RESPONSE: Thank you for this suggestion these are now included in table 3.
14) Before presenting results regarding different sociodemographic categories, please provide data on the whole sample.
RESPONSE: Thank you for this comment. While we believe the data represented in totality, albeit in a more nuanced form, we have added a “total” line to the tables so the reader can easily see the total or overall response. Thank you for this suggestion.
15) Why "age" and "class" were not included in further analysis?
RESPONSE: We performed that analysis and did not identify any significant differences, but we have since added this to the results.
16) Table 4-9, percent should be provided next to numbers.
RESPONSE: Thank you for your suggestion, this has been added.
17) Discussion could be written more in depth. It is more presenting the previous studies instead of discussing.
RESPONSE: Thank you for this comment. We have tried to be conservative in approach with regards to the discussion, letting the largely descriptive nature of the study speak for itself without much additional pontification. However, being encouraged by the reviewer we have made some modest, but targeted edits to the discussion.
Per the reviewer's interest in the time delay. This project was part of a student lead senior thesis project. The delay was a result of difficulties communicating and advancing the paper with former students who had moved on in their professional careers or additional schooling.
Reviewer 3 Report
Comments and Suggestions for Authors
The Impact of COVID-19 Health and Safety Measures on the 2 Physical and Mental Health of College Students
line 54 - not necessary to repeat the MVPA full name again
line 56 - if there were few studies, some ref should be added
survey: have some issues regarding the test-retest reliability. Could it be explained how it was done, since the use of not validated questionnaires is a big issue in this paper?
in the table: race/ethnic only 537 students, less that the total sample size. was an error?
the demographic data was not presented to all 549 students, could it be an error or explain the reason why it's not presented
Author Response
REVIEWER#3
The Impact of COVID-19 Health and Safety Measures on the 2 Physical and Mental Health of College Students
We would like thank the reviewer for their constructive feedback.
line 54 - not necessary to repeat the MVPA full name again
RESPONSE: Deleted, thank you.
line 56 - if there were few studies, some ref should be added
RESPONSE: Thank you we have added some additional references, specifically references 9-11 are on College/University students.
survey: have some issues regarding the test-retest reliability. Could it be explained how it was done, since the use of not validated questionnaires is a big issue in this paper?
RESPONSE: We appreciate the reviewers’ concern and the desire for test-retest reliability. However, there was not a validated questionnaire for the current research study, we did however, model our questions after some previously published research namely the publication of our co-authors below. We have added a statement regarding this limitation to the experimental considerations section.
Mel, A.E.; Stenson, M.C. Physical Activity Changes during the Spring 2020 COVID-19 Shutdown in the United States. Transl. J. Am. Coll. Sport. Med. 2021, 6, 1–8, doi:10.1249/TJX.0000000000000176.
in the table: race/ethnic only 537 students, less that the total sample size. was an error?
RESPONSE: Some respondents were not undergraduate students, of which we excluded. This information has been added to the methods section, and the tables updated.
the demographic data was not presented to all 549 students, could it be an error or explain the reason why it's not presented
RESPONSE: Some respondents were not undergraduate students, of which we excluded. This information has been added to the methods section and sample size been edited accordingly, thank you for allowing us to clarify.
Round 2
Reviewer 2 Report
Comments and Suggestions for Authors
Thank you for your effort to revise the manuscript I still have questions on presentation of intensity and duration and also the table 3. Intensity and duration are gathered as qualitative variables and are better to present as qualitative variables. Also the table 3, the cross tabulation of these two variables is not meaningful.
I still believe that the discussion could be written in depth.
Reviewer 3 Report
Comments and Suggestions for Authors
Even in my previous revision, I did not comment on the discussion, I do agree with the other reviewer that could be improved.